# Development and validation of an artificial intelligence-based pipeline for predicting oral epithelial dysplasia malignant transformation

Adam J. Shephard [1,8], Hanya Mahmood [2,8], Shan E. Ahmed Raza [1], Anna Luíza Damaceno Araújo[3,4], Alan Roger Santos-Silva[5], Marcio Ajudarte Lopes[5], Pablo Agustin Vargas[5], Kris D. McCombe[6], Stephanie G. Craig[6], Jacqueline James [6], Jill Brooks[7], Paul Nankivell [7], Hisham Mehanna[7], Syed Ali Khurram[2,9] & Nasir M. Rajpoot [1,9] ✉

## Abstract

**Background** Oral epithelial dysplasia (OED) is a potentially malignant histopathological diagnosis given to lesions of the oral cavity that are at risk of progression to malignancy. Manual grading of OED is subject to substantial variability and does not reliably predict prognosis, potentially resulting in sub-optimal treatment decisions.
**Method** We developed a Transformer-based artificial intelligence (AI) pipeline for the prediction of malignant transformation from whole-slide images (WSIs) of Haematoxylin and Eosin (H&E) stained OED tissue slides, named ODYN (*Oral Dysplasia Network*). ODYN can simultaneously classify OED and assign a predictive score (ODYN-score) to quantify the risk of malignant transformation. The model was trained on a large cohort using three different scanners (Sheffield, 358 OED WSIs, 105 control WSIs) and externally validated on cases from three independent centres (Birmingham and Belfast, UK, and Piracicaba, Brazil; 108 OED WSIs).
**Results** Model testing yielded an F1-score of 0.96 for classification of dysplastic vs non-dysplastic slides, and an AUROC of 0.73 for malignancy prediction, gaining comparable results to clinical grading systems.
**Conclusions** With further large-scale prospective validation, ODYN promises to offer an objective and reliable solution for assessing OED cases, ultimately improving early detection and treatment of oral cancer.

## Plain language summary

Oral epithelial dysplasia (OED) is a condition where cells in the mouth show abnormal changes that could lead to cancer. The standard method of diagnosis involves looking at a tissue sample (biopsy) under a microscope. However, this method of diagnosis and prediction of cancer risk can be uncertain, resulting in differences in interpretation. In this study, we developed a computer-based tool called "ODYN" to help improve both diagnosis and cancer risk prediction. ODYN examines images of biopsy samples, identifies abnormal areas, and calculates a score that estimates the risk of cancer development. We show that this tool has similar accuracy to the conventional diagnostic criteria, without human involvement. With further testing, ODYN could provide a more objective way to assess OED, helping doctors make better treatment decisions and improving early cancer detection.

Oral epithelial dysplasia (OED) presents a significant challenge in the realm of oral pathology, where accurate diagnosis and early detection are paramount for effective intervention and prevention of malignant progression. OED is a potentially malignant histopathological diagnosis encompassing various lesions of the oral mucosa, typically manifesting as white (leukoplakia), red (erythroplakia) or mixed red-white (erythroleukoplakia) lesions[1,2].

Histopathological grading of Haematoxylin and Eosin (H&E) stained tissue using the World Health Organisation (WHO, 2017[3]) classification system remains the current accepted practice for diagnosis and risk stratification of OED lesions. This is a three-tier system for grading OED into mild, moderate and severe grades based on the presence, severity and location of a wide range of cytological and architectural histological features

(28 in total[4,5]). By its nature, this approach suffers from significant intra- and inter-observer variability and has poor predictive value for malignant transformation risk, potentially impacting on patient management. An alternate binary grading system, categorising lesions as low- or high-risk, based on the number of cytological and architectural features (as listed in the WHO criteria) aimed to improve the reproducibility of grading[6,7]. However, studies have shown significant variability and unreliability in grading using both systems, highlighting the need for more objective and reproducible methods that can better predict malignant transformation risk in OED[8,9].

To address challenges in subjectivity and misclassification of pre-cancerous and cancerous lesions, there is a growing interest in leveraging advanced technologies, particularly deep learning (DL), which has seen extensive use in medical image analysis over the past decade[10–12]. Several state-of-the-art models, such as U-Net[13] and DeepLab[14], have been developed to perform image classification and segmentation. These models typically use convolutional neural networks (CNN), such as ResNet[15], as feature extractors. Within digital pathology, weakly supervised methods have became popular choices for the analysis of histology images, enabling slide-level classification based on patch-level predictions. These methods typically divide WSIs into smaller patches, before using CNNs to extract patch-level features[16–18]. However, despite their success, CNN-based models have limitations such as high computational overhead and difficulty in capturing long-range dependencies in images, when being used for either segmentation or classification.

Transformers have gained widespread attention in recent years as they have been successfully applied in several natural language processing and computer vision tasks such as classification[19–21]. A typical Transformer encoder consists of a multi-head self-attention (MSA) layer, a multi-layer perceptron (MLP), and a layer normalisation (LN). The MSA layer empowers Transformers to capture long-range dependencies, making them a strong candidate for semantic segmentation in medical images[22–24]. Transformers, therefore, have the potential to overcome some of the limitations of traditional CNNs. However, only a handful of methods have applied Transformers for semantic segmentation in medical images[22,25]. Their application in histology has primarily been constrained to classification tasks[26,27], with semantic segmentation left relatively unexplored. This raises the question of whether Transformers can be harnessed for semantic segmentation of histological images.

In this study, we aimed to develop a weakly supervised, DL pipeline that could reliably and objectively segment and classify OED, whilst predicting the risk of malignant transformation in OED, using WSIs of H&E-stained OED slides. Specifically, we achieve this using interpretable nuclear features from dysplastic regions on the WSI. Moreover, we conduct a rigorous evaluation of the performance of our pipeline by comparing it to other state-of-the-art methods. To demonstrate the robustness and generalisability of our approach, we have developed our model using a large cohort with extended validation on unseen datasets acquired from three national and international centres (Birmingham and Belfast, UK, and Piracicaba, Brazil).

## Methods
### Study cohorts
**Development and internal validation cohort.** The training cohort consists of a retrospective sample of histology tissue sections (dating 2008 to 2016, with minimum five-year follow-up data) collected from the Oral and Maxillofacial Pathology archive at the School of Clinical Dentistry, University of Sheffield, UK (referred to as the internal centre, hereafter). During the process of case selection, a Consultant Pathologist (SAK) conducted an initial microscopic inspection of the archived diagnostic slides to confirm the suitability of each case for inclusion. Newly cut 4 μm sections of the selected cases were obtained from the original formalin fixed paraffin embedded blocks and stained with H&E for analysis. The collection, retrieval and staining of sections were conducted between 2020 and 2023 by the same clinicians using standardised protocols, ensuring consistency in slide preparation.

A purposive sampling method was employed, selecting consecutive cases from the pathology archive within the specified time period. In total, 509 slides were collected from 406 patients. The slides were digitised to high-resolution WSIs at 40× objective power using one of three scanners: NanoZoomer S360 (Hamamatsu Photonics, Japan; 0.2258 mpp), Aperio CS2 (Leica Biosystems, Germany; 0.2520 mpp), Pannoramic 1000 (P1000, 3DHISTECH Ltd, Hungary; 0.2426 mpp). Inclusion criteria required sufficient epithelial tissue, high-quality staining, and complete follow-up data (42 cases did not meet these criteria). Exclusion criteria included cases with ulceration, overlying candidal infection, HPV-related OED, or verrucous lesions (based on morphology on H&E). Cases with clinical oral lichen planus (OLP) or coincidental OLP were also excluded. Cases with insufficient tissue, poor staining quality, or incomplete follow-up data were also excluded. Care was taken to ensure a reasonable mix of grades were included.

The resulting cohort comprised 358 WSIs (n = 277 patients) with a confirmed histological diagnosis of OED and 105 WSIs (n = 81 patients) confirmed as non-dysplastic (controls). Due to incomplete follow-up data for five patients with OED (7 WSIs), these cases were only used for algorithm training and excluded from clinical outcome analysis. Thus, the final cohort included 351 WSIs (n = 272 patients) with confirmed diagnosis of OED of which 64 patients (79 WSIs) exhibited malignant transformation. Slides from the same subjects were assigned to the same fold during algorithm training/testing. An overview of the dataset and a CONSORT diagram are given in the Supplementary Information (Table S1 and Fig. S1, respectively).

Clinical follow-up data for the OED cohort included patient age (at time of diagnosis), sex, intraoral site, OED grade (using binary and WHO 2017 systems) and transformation status. Transformation was defined as the progression of a dysplastic lesion to OSCC at the same clinical site within the follow-up period, and transformation time was measured in months. To ensure diagnostic consistency, all cases were evaluated by at least two certified pathologists (PMS, PMF, DJB, KH), who provided an initial diagnosis based on the WHO grading system (between 2008–2016). To confirm the WHO (2017) grade and assign binary grades, the cases were blindly re-evaluated by SAK and a clinician with a specialist interest and expertise in OED analysis (HM).

Amongst the 358 OED WSIs, HM exhaustively delineated regions of interest (ROI) representative of dysplasia in a large subset of 260 OED WSIs, using in-built annotation tools in the QuPath® software[28]. Of the 105 non-dysplastic control WSIs, HM additionally manually delineated the entire epithelium in a subset of 96 WSIs[28].

**Independent validation cohorts.** The ODYN model was tested on three external datasets acquired from:
  i. Precision Medicine Centre, Patrick G. Johnston Centre for Cancer Research, Queen's University Belfast, UK (47 WSIs)
  ii. Institute of Head and Neck Studies and Education, Institute of Cancer and Genomic Sciences, University of Birmingham, UK (42 WSIs)
  iii. Oral Diagnosis Department, Semiology and Oral Pathology Areas, Piracicaba Dental School University of Campinas (UNICAMP), São Paulo, Brazil (19 WSIs)

Owing to the limited size of these datasets, we combined them into a single multi-institutional external test set. Prior to the inclusion of external cases in the study, all WSIs were checked for suitability. Slides of poor quality, insufficient epithelium and cases positive for Candida Albicans or suggestive of Human Papilloma Virus infection were excluded. The WSI cohorts from Birmingham and Belfast were scanned at 40× objective power using a Pannoramic 250 (P250, 3DHISTECH Ltd., Hungary; 0.1394 mpp) and Aperio AT2 (Leica Biosystems, Germany; 0.2529 mpp) whole-slide scanner, respectively, to obtain digital WSIs. The Piracicaba cases were scanned at 20× objective power by an Aperio CS (Leica Biosystems, Germany; 0.4928 mpp) scanner. The same clinical follow-up information was collected as that for the development/internal cohort. The external dataset did not include any control cases. Due to incomplete follow-up data for three patients with OED (3 WSIs), these cases were only used for algorithm

validation and excluded from clinical outcome analysis. Thus, the final cohort included 105 WSIs (from 105 patients), amongst which 44 patients (44 WSIs) exhibited malignant transformation. A summary of this cohort and a CONSORT diagram are provided in the Supplementary Information (Table S1 and Fig. S1, respectively). For model training, HM exhaustively delineated ROIs of dysplasia in 30 cases each from both Birmingham and Belfast, and an additional 18 cases from Piracicaba, using the QuPath® software.

**Inclusion and ethics statement.** Ethical approval for the study was obtained from the NHS Health Research Authority West Midlands (18/WM/0335), and experiments were conducted in compliance with the Declaration of Helsinki. Data collected was fully anonymised. Written consent was not required as data was collected from surplus archived tissue.

### Analytical workflow

**Dysplasia segmentation.** Since dysplastic changes may not be widespread across the entire tissue section in a slide, the first step of developing the DL pipeline involved identification and localisation of the dysplastic tissue regions for semantic segmentation. To achieve this, we trained a Transformer, based on Trans-UNet[22], to automatically detect and segment the different dysplastic regions in each WSI across the training dataset. The model processes input images of size $512 \times 512$ (at 1.0 micron per pixel, mpp, resolution) and outputs a dysplasia segmentation map. Manually annotated ROIs were used as ground truth during training, focusing on areas with confirmed dysplasia in OED cases and the entire epithelium in non-dysplastic controls. These large ROIs typically spanned the entire tissue section in a slide, encompassing both annotated dysplastic epithelium and normal epithelium where present.

For internal model testing, the dataset was split at 80/20, and controlled for both scanner type and OED grade. This resulted in 206 OED and 75 non-dysplastic control WSIs in the training set, and 54 OED and 21 non-dysplastic control WSIs in the internal testing set, with ground truth annotations. Note, a higher proportion of controls were kept in the test set to ensure high specificity of OED segmentation in the non-dysplastic control sample. After tessellating the WSIs and region masks into smaller patches ($512 \times 512$ pixels, 184 pixels overlap, $10\times$ magnification, 1.0 mpp), a total of 19,063 OED and 11,756 non-dysplastic patches were generated for model training/validation on the internal discovery cohort. This totalled 6,341 patches with ground truth annotations from the 78 WSIs in the external cohort. Various stain augmentation algorithms were tested during the development of the final model, using the TIAToolbox[29].

**OED Classification.** A pretrained CNN-based HoVer-Net+[30,31] model was used to segment the epithelium and the individual nuclei across each WSI. To classify OED, the proportion of the epithelium mask (generated by HoVer-Net+) that was segmented as dysplastic (using Trans-UNet) was calculated. This proportion, referred to as the dysplasia-epithelium ratio ($R_{Epith}$), was used to classify slides as dysplastic vs. non-dysplastic, based on an empirically determined threshold.

The threshold for $R_{Epith}$ was selected based on its ability to achieve the highest classification performance (measured by F1-score and AUROC) on the training set of 281 WSIs used for training the Trans-UNet dysplasia segmentation model. This threshold was subsequently validated, internally on the remaining 182 WSIs from Sheffield, and externally on all 108 WSIs. For transparency, the distribution of $R_{Epith}$ values across different OED grades and transformation outcomes was analysed, and boxplots were generated to illustrate these distributions.

HoVer-Net+ was used solely for inference in this task and was not further fine-tuned, given its state-of-the-art performance in epithelium and nuclear segmentation and classification. The model has been extensively pre-trained on OED data[30,31], which ensured its robustness and reliability for this application.

**Malignant transformation prediction (ODYN-scoring).** The WSIs were tessellated into smaller patches ($512 \times 512$ pixels, with 256 pixels overlap at 0.5 mpp) using tissue in the dysplastic regions alone. The nuclear segmentations from HoVer-Net+ were used to generate a total of 168 nuclear-based morphological and spatial features for each (dysplastic) patch. See the Supplementary Information, pp 7, for a list of the features used. These patch-level features were used as input to an MLP to calculate a risk-score for malignant transformation (ODYN-score). Thus, the ODYN-score indicated whether the algorithm predicted the case to have transformed (high-risk) or not transformed (low-risk). The MLP model had three layers with 168 nodes in the input layer, 64 nodes in the hidden layer, and 2 nodes in the output layer. It used a leaky ReLU activation function and dropout (0.2) after the hidden layer. The MLP was trained by Monte Carlo iterative-draw-and-rank sampling (IDaRS[16]), using a symmetric cross-entropy loss function and the Adam optimiser. This loss function was chosen as it has been shown previously to help overcome errors associated with weak labels[16,32]. IDaRS sampling was performed with parameter values of $k = 5$ for top predictive patches and $r = 45$ random patches, using a batch size of 256. On inference, the trained MLP calculated a prediction score for each patch in the dysplastic regions of the WSI, which can be considered the likelihood of a tile belonging to the positive class in the classification task (i.e. transformation). Slide-level scores were then obtained by taking the average prediction score across the top 50% ranked tiles. We used nuclear features with the aim of making the model interpretable. However, we additionally provided comparison to a ResNet34 classifier (trained with Macenko stain augmentation), using deep features, to show the impact on performance (see Supplementary Information, Table S3).

### Statistics and reproducibility

For the evaluation of OED segmentation, on both internal and external testing, large ROIs centred on the annotated tissue section were generated. Dysplasia segmentation performance (aggregated across all ROIs) was measured by calculating the F1-score, Recall and Precision. For internal testing of controls, a single measure of specificity for OED segmentation was reported, since a single incorrectly predicted pixel (e.g. incorrectly predicted as OED), would result in an F1-score, Recall, and Precision values of 0; thus, not giving an accurate representation of the model performance. For the evaluation of OED classification (dysplastic vs non-dysplastic) the F1-score, Recall, and Precision across all slides were measured. An area under the receiving operating characteristic (AUROC) score was also calculated for internal testing.

We used repeated five-fold cross-validation in our ODYN-scoring internal experiments based on the internal cohort. For each fold of cross-validation, we held one fold back for testing, and used the remaining four folds with a 90/10 split of data for training/validation. Experiments were repeated three times with random seeds. We then tested our model externally, by evaluating each model from internal cross-validation (i.e. all 15 folds) on the external data, and ensembling their predictions.

For the evaluation of the ODYN-scoring pipeline, we provide an AUROC score and an area under the precision-recall curve (AUPRC) score across all slides. Survival analyses were additionally conducted to assess the prognostic significance of the ODYN-score in predicting transformation-free survival. Kaplan-Meier curves were generated, and log-rank tests were used to determine the statistical significance of grading (for ODYN-score, WHO and binary grades). We used concordance index (C-index) to measure the rank correlation between predicted risk scores and patients' survival time. The reported C-index is the mean over each repeat of the experiment, whilst the $p$-value is calculated by two times the median $p$-value (from the log-rank test) over all repeats, to get a conservative estimate. A multivariate Cox proportional hazards model was employed, incorporating the ODYN-score, sex and age (and lesion site for internal testing), to predict transformation-free survival. We additionally performed this analysis using the binary and WHO grades in place of the ODYN-score for further comparison. Transformations were right censored at eight years across these

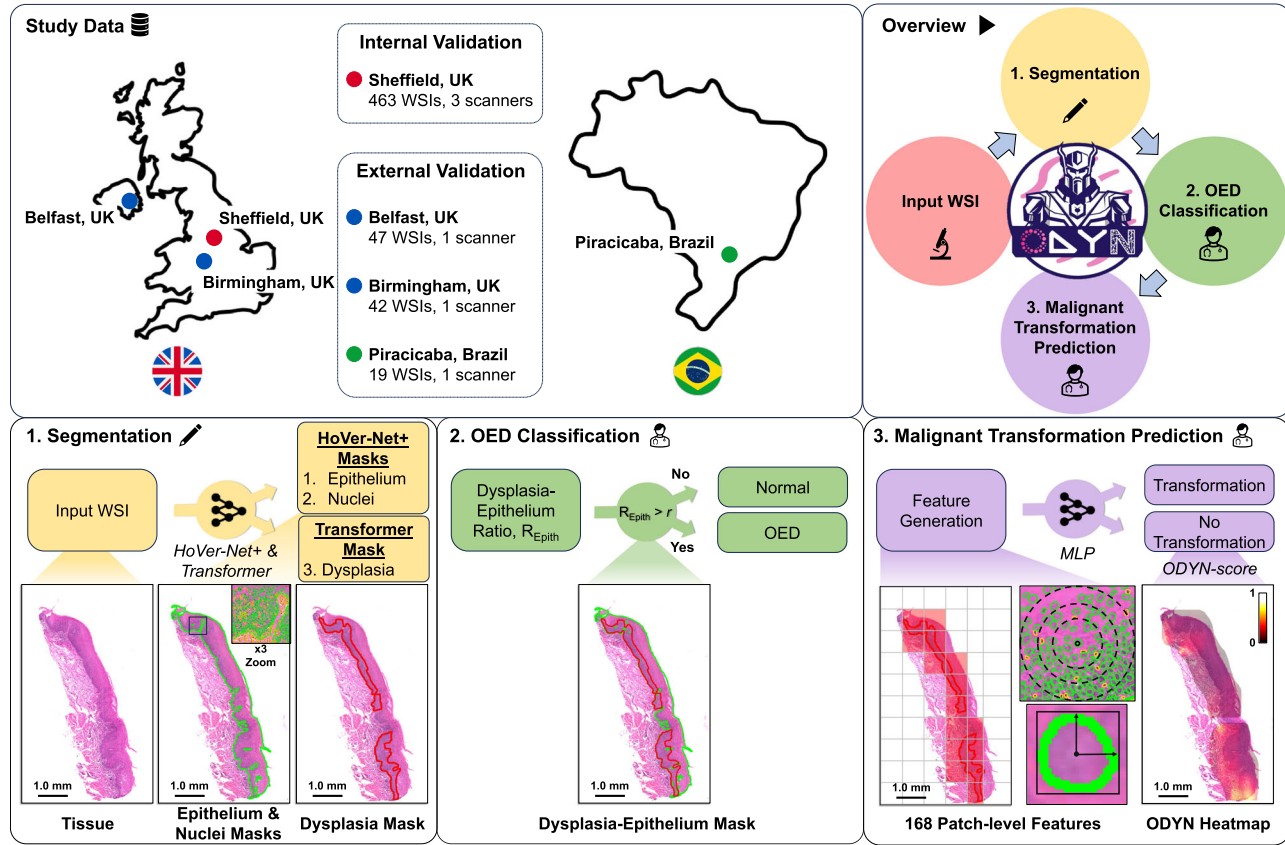

**Fig. 1 | Overview of the ODYN pipeline.** The top left panel shows the study data, whilst the top right panel shows an overview of the ODYN pipeline. The first stage (bottom left) takes an input oral tissue WSI and segments various tissue/cell types. This is done via HoVer-Net+ for epithelial and nuclei segmentation, and Trans-UNet to locate the dysplastic areas of the slide. The second step (bottom middle) diagnoses the input tissue as OED or normal by calculating the ratio of the epithelium that is predicted to be dysplastic. If this is above a threshold (found on model training), then the slide is classified as OED. Finally, the third stage (bottom right) gives a prognosis, i.e. predicts whether the case will become cancerous. To do this, we generate patch-level nuclear features within the dysplastic regions alone and use these within a multi-layer perception (MLP) to predict malignant transformation.

analyses to ensure consistency between internal and external cohorts. We used the hazard ratio (HR) and *p*-value output from the multivariate analyses as further metrics for evaluation. For reporting, we focus on the *p*-value from the multivariate analyses, being a more conservative and robust estimate. However, for completeness we also provide the log-rank *p*-value with the Kaplan-Meier curves.

Finally, we generated nuclear counts and area ratios in the ten top-ranked tiles (as correctly predicted by iterative draw and rank sampling for ODYN-scoring). For nuclear counts, we studied dysplastic epithelial nuclei, normal epithelial nuclei, 'other' nuclei from within the epithelium (i.e. intra-epithelial lymphocytes, IELs), and 'other' nuclei outside the epithelium (i.e. peri-epithelial lymphocytes, PELs). For area ratios, we studied the ratio of the patch that was 'other' tissue, dysplastic epithelium, and normal epithelium. We used Shapiro-Wilk tests to check for normality in counts/areas in each analyses. We then performed two-tail Mann-Whitney U tests (with false discovery rate, FDR correction for multiple comparisons) between patches from cases that ODYN correctly predicted to transform vs does not transform, to determine statistical significance. We additionally calculated effect sizes for these tests (rank-biserial correlation coefficient $r_{rb}$).

### Reporting summary
Further information on research design is available in the Nature Portfolio Reporting Summary linked to this article.

## Results
In this retrospective multi-centric study, we propose an innovative weakly supervised method for predicting the progression of OED lesions to malignancy. We additionally aimed to produce a model that classifies oral tissue slides as being dysplastic vs non-dysplastic. We achieved this by analysing H&E-stained WSIs obtained from oral tissue biopsies, using a CNN, a Transformer and an MLP, in what we have called our Oral DYsplasia Network, 'ODYN' (see Fig. 1).

### Dysplasia segmentation
In many cases of OED, histological atypia is not present across the entire tissue section, and thus, the first step of this work was to identify only the regions where dysplastic changes were present. We trained a Transformer (based on Trans-UNet[22]) to detect and segment the different dysplastic areas in each WSI. Internal testing of the ODYN dysplasia segmentation model demonstrated an F1-score of 0.81 (Recall = 0.85, Precision = 0.77) on OED cases and a specificity of 1.00 on non-dysplastic controls. On external testing, the ODYN model achieved a F1-score of 0.71 (Recall = 0.76, Precision = 0.66). Further, stain augmentation (ODYN-SA, in Supplementary Information, Table S2) did not improve model performance. The results of the ODYN model were superior to that of other state-of-the-art methods including U-Net[13], HoVer-Net+[30,31], DeepLabV3+[33], Efficient-UNet[34], and Swin-UNet[25] (see Supplementary Information, Table S2). Examples of dysplasia segmentation heatmaps are shown in Fig. 2.

### OED classification
Next, we used a pretrained CNN, HoVer-Net+[30,31], to simultaneously segment the epithelium and segment/classify nuclear instances in WSIs. For OED classification, we calculate the proportion of the epithelium mask (output from HoVer-Net+) that was segmented as dysplastic (output from

**Fig. 2 | Dysplasia segmentation heatmap using the ODYN model. a** Severe OED (binary grade: high-risk) which transformed; **b** Mild OED (binary grade: low-risk) which did not transform. The green line depicts the ground truth dysplasia segmentation.

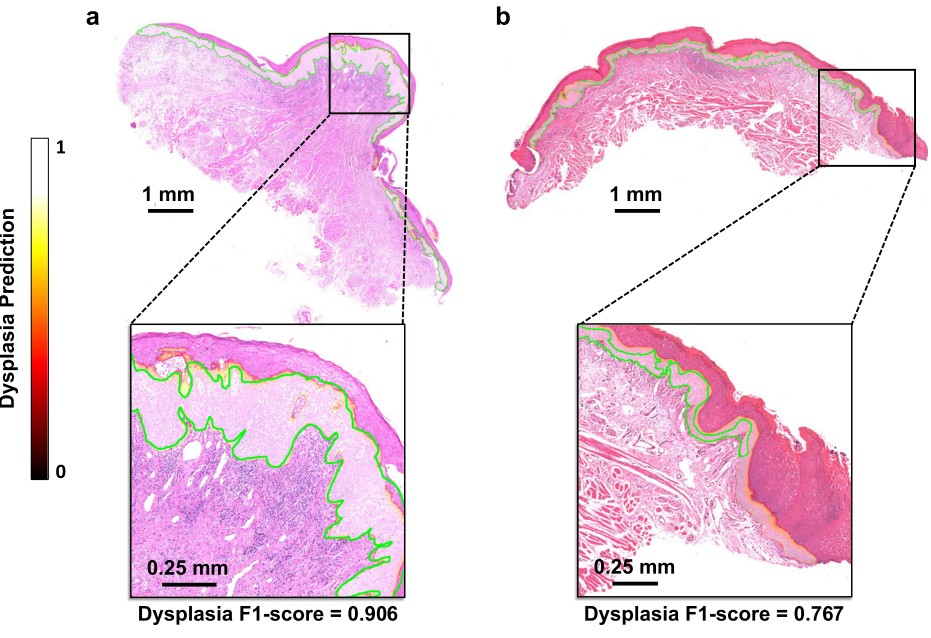

Dysplasia F1-score = 0.906          Dysplasia F1-score = 0.767

the dysplasia segmentation Trans-UNet model). We used an empirically determined threshold to classify slides as being dysplastic vs. non-dysplastic. On internal testing, we achieved an F1-score of 0.96 (AUROC = 0.93, Recall = 0.94, Precision = 0.97). The performance remained high on external testing, gaining an F1-score = 0.96 (Recall = 0.93, Precision = 1.00), showing the robustness and generalisability of the proposed method.

To further explore the variability of $R_{Epith}$, we analysed its distribution across OED grades and transformation outcomes. Boxplots illustrating these distributions can be seen in Fig. 3, providing additional insights into how this threshold correlates with prognostic outcomes. Shapiro-Wilk tests found the score to be not normally distributed across internal ($p < 0.001$) and external ($p = 0.01$) testing. To compare these scores across cases, for transformation status and binary grade, we used non-parametric Mann-Whitney U tests with rank-biserial correlation coefficient $r_{rb}$, as effect size. For the WHO grade, we used Spearman's correlation $\rho$, with $p$-values calculated via permutation tests. Unless otherwise specified, all continuous variables are reported as medians (M) with interquartile ranges (IQR).

On internal testing, we found the $R_{Epith}$ to be significantly associated with transformation (non-transformed: M = 0.26 (IQR = 0.17–0.35); transformed: 0.39 (0.24–0.55); $r_{rb} = 0.34$, $p < 0.001$), binary grade (low-risk: 0.24 (0.15–0.33); high-risk: 0.36 (0.26–0.52); $r_{rb} = 0.42$, $p < 0.001$), and WHO grade ($\rho = 0.44$, $p < 0.001$). Similarly on external testing, $R_{Epith}$ was significantly associated with transformation (non-transformed: 0.20 (0.15–0.32); transformed: 0.35 (0.22–0.45); $r_{rb} = 0.37$, $p = 0.001$), binary grade (low-risk: 0.19 (0.13–0.31); high-risk: 0.32 (0.18–0.44); $r_{rb} = 0.36$, $p = 0.002$), and WHO grade ($\rho = 0.31$, $p < 0.001$).

**Malignant transformation prediction**
We generated patch-level morphological features in the dysplastic regions of OED cases, which were used as input to an MLP to calculate a risk-score for malignancy progression (the ODYN-score). On internal cross-validation, we attained an AUROC of 0.71 for predicting malignant transformation, which remained relatively constant on external validation, rising to 0.73 (see Table 1). These scores are competitive to existing clinical grading systems including WHO (2017) and binary grades. However, it must be noted that the binary grading system had a higher AUPRC of 0.72 when compared to the ODYN-score. For a complete evaluation, we also compared our ODYN-

score to the other grading systems through a survival analysis (see Fig. 4). On internal testing, our ODYN-score gained a comparable C-index of 0.66 and hazard ratio of 3.86, when compared to the other grading systems, and was shown to be significant ($p < 0.001$). On external testing, the ODYN-score (C-index = 0.63) again attained comparable performance to both the binary grading system (C-index = 0.62) and WHO grading system (G1 stratification; C-index = 0.61), with all three being significant. The ODYN-score continues to surpass the WHO G2 stratification in terms of C-index and hazard ratio on both internal and external testing. Overall, these results show the prognostic significance and utility of the ODYN-score, being comparable to that of a pathologist's binary grade for predicting transformation-free survival.

**Feature analysis**
For our feature analysis, we compared both nuclear counts and area ratio in the top ten patches from cases that ODYN predicted to transform (i.e. true positives, TPs) against those correctly predicted to not transform (i.e. true negatives, TNs). This analysis was performed on the external data alone, and boxplots are given in the Supplementary Information, Fig. S2. The nuclear counts and area ratios were found to not be normally distributed by Shapiro-Wilk tests (all $p < 0.001$), and thus we used non-parametric Mann-Whitney U tests in the following analyses with a rank-biserial correlation coefficient $r_{rb}$ effect size. Unless otherwise specified, all continuous variables are reported as medians (M) with interquartile ranges (IQR).

The nuclear count analysis found a significantly higher number of 'other' nuclei within the non-epithelial tissue (TN: M = 56 (IQR = 24–104); TP: 183 (93–254); $r_{rb} = 0.63$, $p < 0.001$), in TPs when compared to TNs. It also showed a significantly higher number of 'other' nuclei within the epithelium (i.e. intra-epithelial lymphocytes, IELs) in TNs when compared to TPs, however with a weak effect size (TN: 16 (8–29); TP: 8 (0–25); $r_{rb} = -0.27$, $p < 0.001$). It also displayed a significantly higher number of both dysplastic epithelial nuclei (TN: 128 (85–163); TP: 29 (0–94); $r_{rb} = -0.65$, $p < 0.001$) and normal epithelial nuclei (TN: 39 (12–72); TP: 0 (0–17); $r_{rb} = -0.63$, $p < 0.001$) within TNs when compared to TPs. The area ratio found a significantly higher number of 'other' tissue in TPs when compared to TNs (TN: 0.13 (0.03–0.32); TP: 0.63 (0.20–0.97); $r_{rb} = 0.59$, $p < 0.001$). Finally, it also showed a significantly higher number of

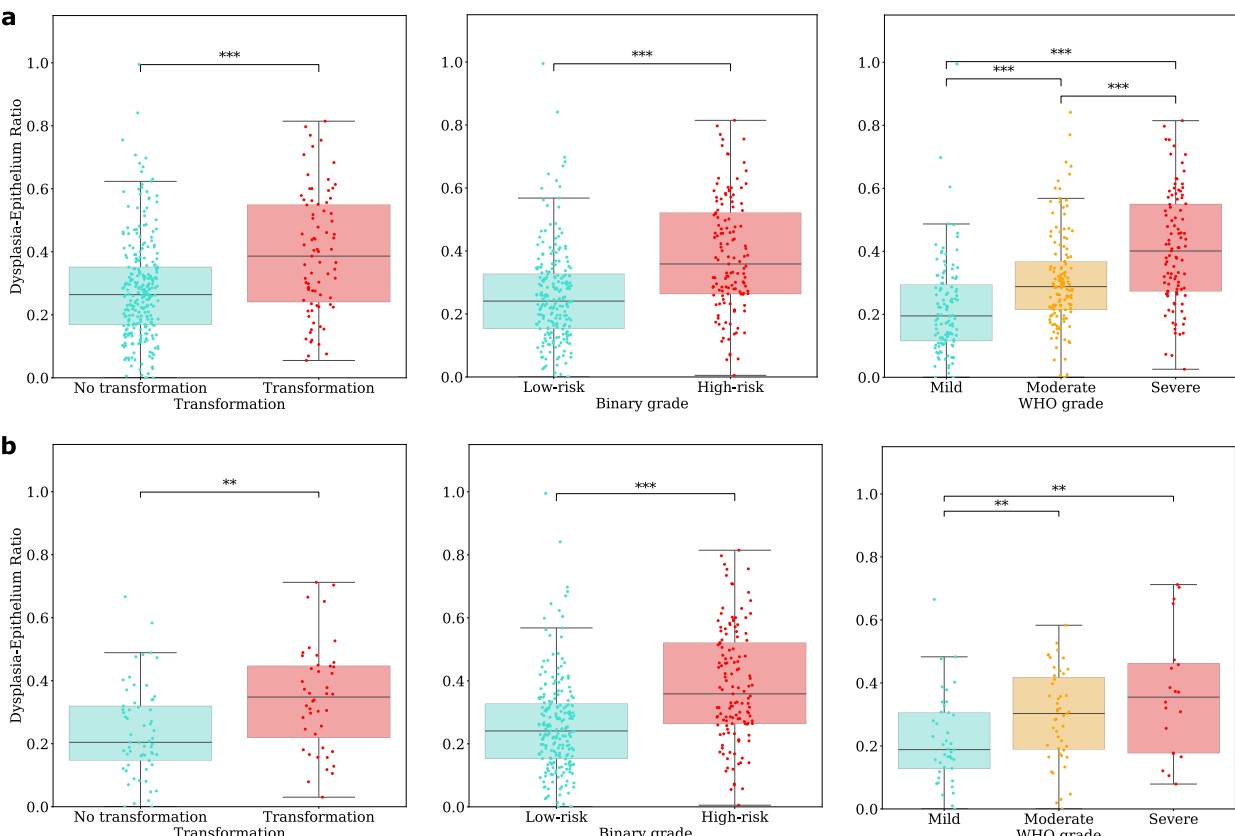

**Fig. 3 | The distribution of dysplasia-epithelium ratios across OED cases based on transformation and grade.** Boxplots showing the distribution of dysplasia-epithelium ratios in OED cases according to: transformation status (left), where transforming cases are red and not transforming are cyan; binary grade (middle), where low-risk cases are cyan and high-risk cases are red; and WHO grade (right), where mild cases are cyan, moderate orange, and severe are red. The top row (**a**) is for internal testing and the bottom row (**b**) is for external testing.

both dysplastic epithelium (TN: 0.15 (0.05–0.26); TP: 0.04 (0.00–0.17); $r_{rb} = -0.36$, $p < 0.001$) and normal epithelium (TN: 0.36 (0.19–0.56); TP: 0.05 (0.00–0.36); $r_{rb} = -0.49$, $p < 0.001$) within TNs compared to TPs.

## Discussion

Several studies have explored the application of machine learning, including DL, to study OED. The general focus of these methods has been to segment the epithelium (and the nuclei), either manually or via DL models[30,35,36]. These segmentations have then been used in further DL models to predict grade or transformation[31,35,37] or for pathologist-curated features based on digital images[38]. However, no previous studies have fully integrated segmentation of dysplastic regions into a unified pipeline to further classify OED cases and predict their malignant transformation.

In this study, we introduce ODYN, a Transformer-based pipeline that integrates OED segmentation, classification and malignant transformation prediction into a single automated framework. Unlike previous studies, which focus on individual tasks such as segmentation or transformation prediction, ODYN combines these steps into a unified workflow. This pipeline has been developed using the largest and most diverse multicentric OED dataset to date, digitised using six different scanners. The results obtained through rigorous testing and validation demonstrate the effectiveness of our models in various aspects of OED analysis. We highlight that ODYN is the first model to specifically focus on dysplasia segmentation for downstream prediction of malignant transformation, a key clinical outcome in OED.

The ODYN dysplasia segmentation performance has consistently outperformed other state-of-the-art DL models. We found only one other

study to attempt dysplasia segmentation in OED[36]. The authors used a DeepLabV3+ model and evaluated it at the patch level on moderate/severe cases from a single centre. Our study improved on this, using a new Transformer-based architecture evaluated at the WSI-level on all types of OED (mild, moderate and severe) from multiple centres, gaining higher F1-scores. Furthermore, the ODYN model has demonstrated good generalisability across external unseen datasets, indicating its robustness and applicability in diverse clinical settings. This highlights the potential of Transformer-based architectures in accurately delineating regions of dysplasia in H&E-stained WSIs of oral epithelial tissue and reinforces the clinical value of ODYN's unified pipeline. However, while non-dysplastic controls were included in internal testing to comprehensively assess the dysplasia detector's performance, non-dysplastic control cases were unavailable for external testing, and we acknowledge this as a limitation of our study. Despite this, ODYN's ground-breaking approach has the potential to redefine the landscape of OED diagnosis by providing more precise and consistent results.

ODYN has also demonstrated promising results for OED classification. In this study, we used the predicted dysplastic proportion of the epithelium in a WSI to determine a diagnosis of OED. We chose this method to classify a WSI as dysplastic, rather than classifying a WSI as dysplastic solely based on the presence of any predicted dysplasia. We made this choice because our model predictions often included small areas of false positives. This decision to define a threshold proved to be successful on both internal and external testing. The high precision and recall achieved in classifying OED indicate the potential for automated diagnosis, which has the potential to increase diagnostic efficiency. Moreover, the dysplasia-epithelium ratio

**Table 1 | Slide-level results for transformation prediction**

| Model | Internal Validation | | | | | External Validation | | | | |
|---|---|---|---|---|---|---|---|---|---|---|
| | AUROC | AUPRC | HR | p | C-Index | AUROC | AUPRC | HR | p | C-Index |
| WHO grade G1 | 0.67 (0.03) | **0.63 (0.07)** | **9.16 [3.68–22.80]** | <0.001 | 0.66 (0.00) | 0.65 (0.00) | **0.72 (0.00)** | 2.43 [1.12–5.29] | 0.025 | 0.61 (0.00) |
| WHO grade G2 | 0.61 (0.06) | 0.47 (0.11) | 2.71 [1.73–4.26] | <0.001 | 0.62 (0.00) | 0.57 (0.00) | 0.59 (0.00) | 1.62 [0.82–3.19] | 0.164 | 0.56 (0.00) |
| Binary grade | **0.73 (0.05)** | 0.62 (0.07) | 6.03 [3.63–10.01] | <0.001 | **0.71 (0.00)** | 0.68 (0.00) | **0.72 (0.00)** | 2.84 [1.36–5.92] | 0.005 | 0.62 (0.00) |
| ODYN-score | 0.71 (0.07) | 0.43 (0.12) | 3.86 [2.04–7.69] | <0.001 | 0.66 (0.01) | **0.73 (0.05)** | 0.67 (0.05) | **2.95 [1.44–6.02]** | **0.003** | **0.63 (0.04)** |

Best values in bold.
Here, WHO grade G1 is mild vs moderate/severe cases, whilst WHO grade G2 is mild/moderate vs severe cases. For AUROC, AUPRC and C-Index, the mean value is given with the standard deviation in brackets. For the hazard ratio, HR, we additionally provide the 95% confidence interval in square brackets.

($R_{Epith}$) showed strong correlations with clinically relevant outcomes, including OED grade and transformation status. This highlights its potential not only as a diagnostic tool but also as a prognostic biomarker, further underscoring the utility of ODYN in clinical practice.

The application of ODYN-produced segmentation maps in predicting malignant transformation represents a significant advancement in computational pathology. Notably, this approach outperforms the *OMTscoring* pipeline proposed by Shephard et al.[31] with a substantial improvement in AUROC score (see Supplementary Information, Table S3 for comparative results). However, some comments must be made regarding model performance on external testing. Despite the AUROC and AUPRC remaining high for ODYN, there was a substantial drop in C-index. This drop was also seen for the WHO and binary grades, suggesting that this may be attributed to differences between internal and external datasets (i.e. a domain shift). An analysis of the data used for external testing showed a substantially different transformation-free survival rate for external centres. We see only 23% of cases to transform on internal testing. In contrast, nearly 42% of cases transformed in the external cohorts. This variation in the number of events is a clear indication of a type II prior (domain) shift between internal and external cohorts[39] (see Supplementary Information, Fig. S3, for Kaplan-Meier transformation-free survival curves), and is the clinical reality of retrospective cohorts.

Further, on external validation, low-risk ODYN cases demonstrated a 22% malignant transformation rate, highlighting a potential limitation of the model. While the ODYN-score primarily relies on cytonuclear features of atypia, recent evidence suggests that architectural changes, often overlooked in traditional dysplasia grading, play a critical role in predicting malignant transformation[40,41]. These findings align with reports that lesions with minimal cytonuclear atypia but significant architectural abnormalities can carry a comparable risk of progression as those with pronounced cytonuclear changes. Future models could be enhanced by incorporating architectural features to improve prognostic accuracy.

The provided approach offers a significant level of explainability; a crucial aspect for translating computational models to clinical practice. Our model used morphological/spatial features within (and around) dysplastic areas to generate a prediction, thus emulating the features used by the pathologist in OED grading. Our feature analysis allowed the exploration of different nuclear types within dysplastic vs normal epithelium. These analyses showed, unsurprisingly perhaps, that more dysplastic nuclei were present in the patches that were predicted to transform (vs not transform). Corroborating this, they additionally showed cases that were correctly predicted to not transform to have more normal epithelial tissue (and nuclei). Moreover, cases that transformed exhibited increased 'other' nuclei in the connective tissue. We posit that this elevated density of 'other' nuclei around the epithelium within transforming cases likely indicates the presence of peri-epithelial lymphocytes (PELs). Furthermore, emerging evidence from Bashir et al.[35] highlights a higher density of PELs in cases undergoing malignant transformation. These findings align with previous research, noting increased immune cell infiltration in tongue lesions progressing to OSCC[42] and identifying distinct immune-related subtypes in moderate and severe OED[43].

We believe that the application of cutting-edge DL techniques, such as the ODYN pipeline, has huge translational potential which could help improve the accuracy and objectivity of OED diagnosis and grading. By fully integrating segmentation, classification, and transformation prediction in a single pipeline, ODYN simplifies clinical workflows while providing robust results. In addition to this, AI-based pipelines can improve prognostic reliability for prediction of cancer risk to improve patient outcomes. Future research should explore the scalability of the ODYN model to accommodate a broader range of oral conditions (such as those which can mimic OED) and tissue variations to assess whether it can accurately discriminate OED from other similar appearing conditions whilst still accurately predicting malignancy risk. This will enhance the clinical utility of the model and ultimately help provide more personalised patient care.

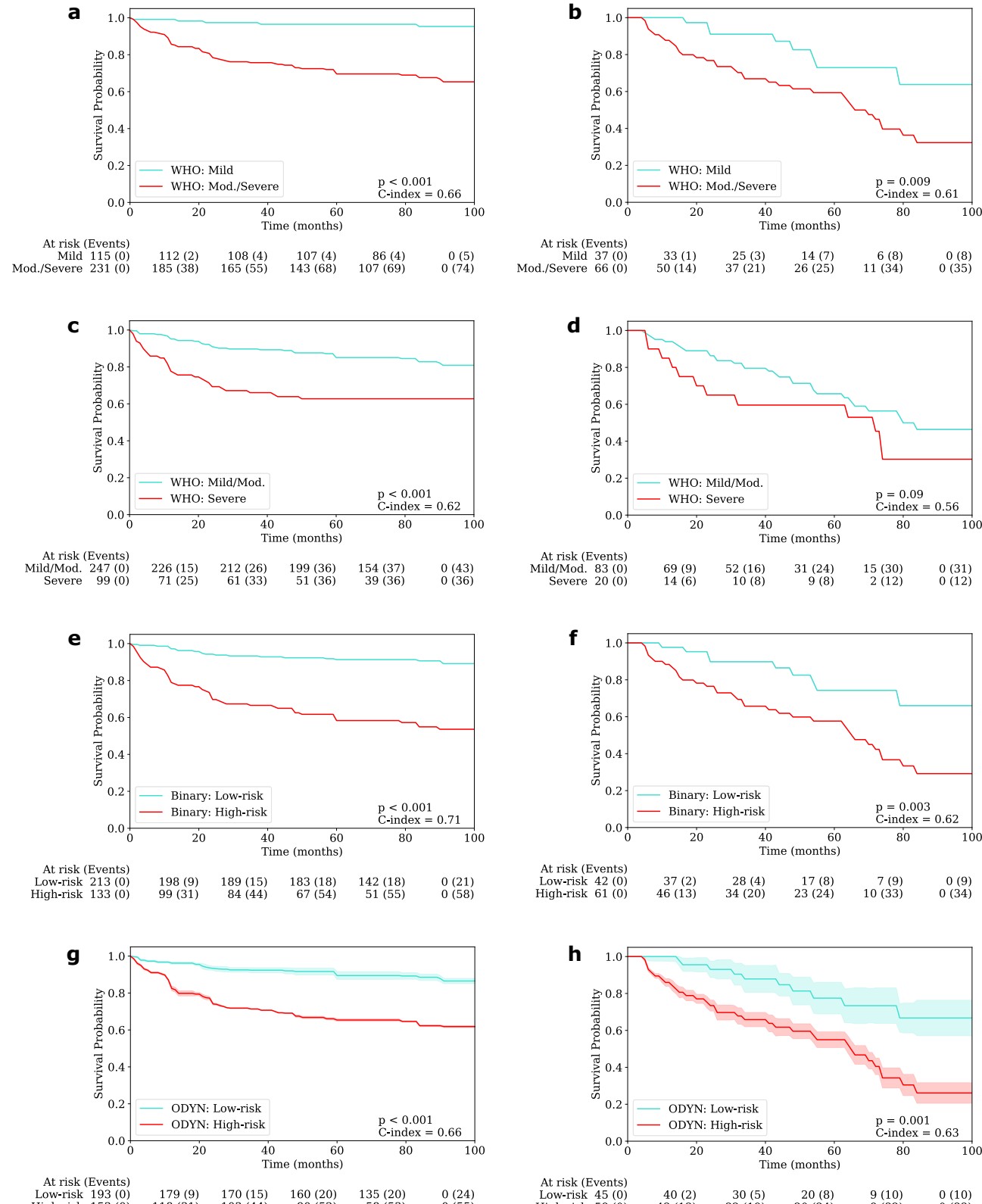

**Fig. 4 | Kaplan-Meier transformation-free survival curves.** Internal testing is on the left and external testing is on the right. The top row (**a**, **b**) is WHO grade G1 (i.e. Mild vs Moderate/Severe OED), second row (**c**, **d**) is WHO grade G2 (i.e. Mild/Moderate vs Severe OED), followed by the Binary grade (**e**, **f**) and the ODYN-score (**g**, **h**). Confidence intervals supplied are generated by the standard deviation of the model output over repeated runs of the experiment. All cases are right censored at eight years (96 months).

The authors acknowledge challenges and opportunities for future research based on this study. A potential challenge highlighted by this work is the need to address the interpretability of DL models in clinical practice. We have therefore used an interpretable model for transformation prediction that considers known histological features (e.g. shape and size variations of nuclei) to generate predictions from dysplastic ROIs. We provide heatmaps for each slide to help explain model decisions. We believe such approaches can enhance trust and acceptance amongst healthcare professionals.

We acknowledge that strict inclusion criteria were necessary to ensure data quality and reliability for model training. However, we recognise that this approach may limit immediate clinical translation. Future validation in larger and more heterogeneous datasets, including cases with minor artefacts is required. Future work could also explore the incorporation of automated methods to identify and manage such issues, potentially reducing attrition while preserving data quality. These steps will help address the balance between ensuring robust model training and achieving broader clinical applicability.

The authors additionally acknowledge limitations related to the retrospective nature of the study. It would have been of interest to further explore the model performance for predicting OED recurrence. However, as there is no standardised treatment protocol for OED, there may have been variations in patient management between centres, and it is also difficult to reliably know the difference between true recurrence and field change. We would have additionally liked to incorporate social risk factors (e.g. smoking, alcohol consumption) in the multivariable modelling, however, it was not possible to acquire consistent information between the different centres. These issues could be addressed by a future prospective validation study. Despite this, the external validation of our models across multiple centres and scanners is a notable strength of this study. Future research could explore the application of ODYN in even more diverse clinical settings and expand its utility to other histopathological tasks beyond OED analysis. We suggest testing the method on other head and neck precancerous lesions, such as laryngeal dysplasia, as an interesting future direction of research.

In conclusion, our study signifies a substantial leap forward in the field of digital oral pathology, offering a powerful tool in ODYN for the detection, segmentation, and classification of OED, which we have made publicly available. This technology, underpinned by DL and Transformer-based architectures, showcases the potential of computational pathology to revolutionise the diagnosis and management of OED. The model's exceptional performance in both internal and external testing, along with its ability to improve transformation prediction, underscores its potential to impact clinical practice positively. By addressing challenges and continuing to refine the model, we envision ODYN playing an important role in improving the diagnosis and management of OED and potentially other head and neck precancerous lesions in the future.

## Data availability
We are unable to share the whole slide images and clinical data due to restrictions in the ethics applications. The source data for Fig. 3 is found within Supplementary Data 1, and for Fig. 4 is in Supplementary Data 2.

## Code availability
In the spirit of reproducibility, we have made the inference code for our pipeline available online, with model weights[44].

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

## Acknowledgements

This study was supported by a Cancer Research UK Early Detection Project Grant, as part of the ANTICIPATE study (grant no. C63489/A29674) in addition to funding from the National Institute for Health Research (award no. NIHR300904). ALDA was funded by The São Paulo Research Foundation (grant no. 2021/14585-7). The authors express their gratitude to Professor Paul Speight (PMS), Professor Paula Farthing (PMF), Dr Daniel Brierley (DJB), and Professor Keith Hunter (KH) for their valuable contribution in providing the initial histological diagnoses. The authors would additionally like to thank Dr Mark Eastwood for his help with the visualisation of cases (and their segmentations) on the tiademos server (https://tiademos.dcs.warwick.ac.uk/).

## Author contributions

A.S., Hanya M., S.E.A.R., S.A.K. and N.M.R. designed the study with the help of all co-authors. A.S., Hanya M. and N.M.R. developed the computational methods. A.S. wrote the code and carried out all the experiments. Hanya M. and S.A.K. provided the WSI annotations. S.A.K. and Hanya M. obtained the ethical approval and retrieved the histological and clinical data from Sheffield. K.M., S.C. and J.J. contributed to the collection of the histological and clinical data from Belfast. J.B., P.N. and Hisham M. contributed to the collection of the histological and clinical data from Birmingham. A.L.D.A., A.R.S.S., M.A.L., P.A.V. contributed to the collection of the histological and clinical data from Piracicaba, Brazil. All authors contributed to the writing of the manuscript.

## Competing interests

N.M.R. is the co-founder, CEO and CSO, and a shareholder of Histofy Ltd. He is also the GSK Chair of Computational Pathology and is in receipt of research funding from GSK and AstraZeneca. S.A.K. is a shareholder of Histofy Ltd. All other authors have no competing interests to declare.

## Additional information

[1]Tissue Image Analytics Centre, Department of Computer Science, University of Warwick, Coventry, UK. [2]School of Clinical Dentistry, University of Sheffield, Sheffield, UK. [3]Head and Neck Surgery Department and LIM 28, University of São Paulo Medical School, São Paulo, State of São Paulo, Brazil. [4]Hospital Israelita Albert Einstein, São Paulo, State of São Paulo, Brazil. [5]Faculdade de Odontologia de Piracicaba, Universidade Estadual de Campinas (FOP-UNICAMP), Piracicaba, State of São Paulo, Brazil. [6]Precision Medicine Centre, Patrick G. Johnston Centre for Cancer Research, Queen's University Belfast, Belfast, UK. [7]Institute of Head and Neck Studies and Education, Institute of Cancer and Genomic Sciences, University of Birmingham, Birmingham, UK. [8]These authors contributed equally: Adam J. Shephard, Hanya Mahmood. [9]These authors jointly supervised this work: Syed Ali Khurram, Nasir M. Rajpoot. ✉e-mail: n.m.rajpoot@warwick.ac.uk

