## [Transparent Peer Review file · Communications Medicine]

Development and Validation of an Artificial Intelligence-based Pipeline for Predicting Oral Epithelial Dysplasia Malignant Transformation

Corresponding Author: Professor Nasir Rajpoot

Version 0:

Reviewer comments:

Reviewer #1

(Remarks to the Author)

Shepard AJ, Mahmood H et al. propose a state-of-the-art deep learning (DL) method combining vision transformers, CNNs, and MLPs to segment, classify, and predict the transformation of oral epithelial dysplasia (OED) lesions. The authors demonstrate the potential of DL in clinical applications, presenting a translational approach. All datasets used are real-world cases, and importantly, the training set from one cohort generalizes and performs well on external datasets across the UK and one centre in Brazil. The study is well-developed, though several concerns need clarification.

1. Complexity of Nuclear Segmentation: While HoverNet+ is a powerful model for nuclear segmentation, this reviewer finds that it adds a layer of complexity. It is sometimes unclear why and how it has been adapted within the pipeline. It would be recommended a detailed schematic as supplementary file.
2. Unclear Segmentation Purpose in Section 2.2: In Section 2.2, the authors mention using HoverNet+ for segmentation. HoverNet was developed primarily to address challenges in nuclear segmentation and cell classification. It is unclear if it was used to segment the epithelium and nuclei specifically. Did the authors intend to segment all nuclei within the epithelium? If so, how were the epithelial masks generated?
3. Empirically Determined Threshold: The authors mention obtaining an "empirically determined threshold." What were the range and distribution of these proportions among cases, and was this threshold already associated with prognosis?
4. Pipeline Novelty: The novelty of the proposed pipeline is not well substantiated, given that previous state-of-the-art models already perform well. The paper's main strength is its evaluation on three external datasets. The discussion includes a statement that "These segmentations have been used in further DL models to predict grade or transformation or for pathologist-curated features based on digital images. However, there has been little focus on segmenting dysplastic regions solely for downstream prediction of malignant transformation." This is somewhat unclear, as the primary clinical outcome is to predict transformation. The statement does not fully justify the added segmentation steps.
5. Case Selection and Staining Clarification: The authors state, "After microscopic inspection of the tissue sections by a Consultant Pathologist (SAK), newly cut 4 μm sections" were used. Does this mean that diagnostic slides were not used for training, but instead fresh, newly stained sections were used? Could the authors clarify if these sections were stained simultaneously or at different times? If at different times, in batches of how many slides per batch?
6. High Case Attrition Due to Artifacts: It is surprising that the initial cohort of 406 patients was reduced to 358 due to artifacts, resulting in a loss of over 11% of cases. For a translational study, this suggests a notable impact on clinical utility. The authors have done prior work on artifact detection; could such approaches be used here to reduce case attrition? Alternatively, could the authors report model accuracy in the presence of artifacts?
7. Utilization of Vision Transformers and ROIs: Although the authors leverage vision transformers within their architecture, they mention feeding the models with regions of interest (ROIs) from a large cohort of 260 slides, with the epithelium delineated in 105 WSIs. This raises questions about the consistency of ROIs in the training data and its impact on model performance.

Minor Comments:

- Figure 1: In the Epithelium/Nuclear Masks image, no nuclear segmentation mask is visible. Is this correct?
- Table S2 (ODYN-SA Model): The supplementary table lists a model, ODYN-SA, which is not referenced in the methods or main text. Could the authors clarify if this is accurate?
- Inclusion of Control Cases: It would strengthen the study to include control cases or OED cases that did not transform, allowing evaluation of dysplasia detector performance in distinguishing non-transforming from transforming OED.
- Heatmap Colormap Selection: Jet colormaps can introduce interpretive bias in heatmaps. I recommend the authors consider this and refer to guidelines here: <https://www.nature.com/articles/s41467-020-19160-7>

Reviewer #2

(Remarks to the Author)

Shephard et al constructed an AI based model to discern histopathological dysplasia in oral epithelium and to link the findings to the chance of malignant transformation. This is an important piece of work, well executed. Since the authors provide the toll they used, independent parties can reproduce the results. The authors used different scanners to scan whole slide images and they used different cohorts from different pathology institutes, thereby correcting for possible institutional related HE features. They use the WHO criteria to grade dysplasia and to divide them into a three tier system of mild, moderate and severe dysplasia. Next to that, the WHO describes 28 features which are indicative for dysplasia. The weakness of the present WHO description of oral epithelial dysplasia lies in the fact that in the three tier system there is a strong emphasis on the cytonuclear atypia of the epithelium and the extent in which this is present in the level of the epithelium. We now know that there are also cases with little cytonuclear changes but mostly architectural changes, that are as prone to develop malignancy as cases with more pronounced cytonuclear changes (Wils LJ, et al. The role of differentiated dysplasia in the prediction of malignant transformation of oral leukoplakia. J Oral Pathol Med. 2023 Nov;52(10):930-938. doi: 10.1111/jop.13483. Epub 2023 Sep 25. PMID: 37749621.; Brouns ER, et al. Oral leukoplakia classification and staging system with incorporation of differentiated dysplasia. Oral Dis. 2023 Oct;29(7):2667-2676. doi: 10.1111/odi.14295. Epub 2022 Jul 7. PMID: 35765231.).

Could the authors further elude to this in the discussion? They find a malignant transformation rate of 40% in the low-risk cases using the ODYN score in the external cohort (fig 3). Could this be due to the fact that the model puts to much emphasis on the cytonuclear features of atypia?

Version 1:

Reviewer comments:

Reviewer #1

(Remarks to the Author)

The authors have done an excellent job addressing the reviewers' comments. I have no additional feedback to provide.

Reviewer #2

(Remarks to the Author)

Upon previous review, Rajpoot et al have thoroughly revised the manuscript, addressing the points made by the reviewers.

Manuscript# COMMSMED-24-1120A

Development and Validation of an Artificial Intelligence-based Pipeline for Predicting Oral Epithelial Dysplasia Malignant Transformation

We express our gratitude to the reviewers for their time and thoughtful feedback on our manuscript. We carefully consider and respond to each comment and suggestion from all the reviewers in the rest of this document. Our response appears in blue font. Within the revised manuscript we have made changes in red font.

Editor:

Your manuscript entitled "Development and Validation of an Artificial Intelligence-based Pipeline for Predicting Oral Epithelial Dysplasia Malignant Transformation" has now been seen by 2 referees. You will see from their comments below that while they find your work of considerable interest, some important points are raised. We are interested in the possibility of publishing your study in Communications Medicine, but would like to consider your response to these concerns in the form of a revised manuscript before we make a final decision on publication.

In particular, we would expect your revision to address Reviewer #3's concerns regarding the focus on cytonuclear features within the model, and the concerns of Reviewer #1 regarding the segmentation tools used.

We therefore invite you to revise and resubmit your manuscript, taking into account the points raised. Please highlight all changes in the manuscript text file.

We thank the editorial team and reviewers for their comments, and thorough feedback. The authorship team have carefully considered the points raised and provide a point-by-point response to each of the reviewer's comments, which we believe has substantially improved the quality and impact of our manuscript.

Reviewer #1:

- 1.1 Complexity of Nuclear Segmentation: While HoverNet+ is a powerful model for nuclear segmentation, this reviewer finds that it adds a layer of complexity. It is sometimes unclear why and how it has been adapted within the pipeline. It would be recommended a detailed schematic as supplementary file.

We thank the reviewer for their comments and acknowledge the extra complexity that it adds to the overall pipeline. We would like to clarify that the trained Trans-UNet model was used to semantically segment areas of dysplasia, while HoVer-Net+ was separately used to segment the epithelium and nuclear instances in the WSIs. These two segmentation tasks serve distinct purposes within the pipeline: (1) epithelium segmentation by HoVer-Net+ was crucial for determining the dysplastic-epithelium

ratio for OED classification, and (2) nuclear segmentations by HoVer-Net+ were necessary for extracting nuclear features for malignant transformation prediction.

To improve clarity, we have updated the relevant sections in the Methods (4.2.2) to include a detailed explanation of these steps. We have added more details to Figure 1 to further show how the models are integrated into the pipeline, along with their distinct outputs.

- 1.2 Unclear Segmentation Purpose in Section 2.2: In Section 2.2, the authors mention using HoverNet+ for segmentation. HoverNet was developed primarily to address challenges in nuclear segmentation and cell classification. It is unclear if it was used to segment the epithelium and nuclei specifically. Did the authors intend to segment all nuclei within the epithelium? If so, how were the epithelial masks generated?

We thank the reviewer for highlighting this important point. We would like to clarify that HoVer-Net+ differs from HoVer-Net in that it performs simultaneous semantic segmentation of the epithelium, and segmentation/classification of nuclear instances, whereas HoVer-Net only did segmentation/classification of nuclear instances. We have made this distinction clearer in the Section 2.2 of the manuscript. We have additionally revised Section 2.2 to clarify that HoVer-Net+ was used specifically to segment the epithelium and nuclei, while the Trans-UNet model was responsible for dysplasia segmentation. The epithelium masks generated by HoVer-Net+ were overlaid with the dysplasia masks from the Trans-UNet model to calculate the dysplastic-epithelium ratio, which served as the key feature for OED classification. These clarifications are included in both the Results (2.2) and Methods (4.2.2) sections.

- 1.3 Empirically Determined Threshold: The authors mention obtaining an “empirically determined threshold.” What were the range and distribution of these proportions among cases, and was this threshold already associated with prognosis?

We thank the reviewer for their insightful feedback. The empirically determined threshold was selected based on its ability to achieve the highest classification performance, as measured by F1-score and AUROC, on the training set used to train the Trans-UNet model for dysplasia segmentation. This threshold was frozen and the model with the frozen threshold value was subsequently validated on both the internal and external testing sets to ensure its generalisability.

To address the reviewer’s query regarding the range and distribution of these proportions, we have conducted an additional analysis to explore how the dysplasia-epithelium ratio (R_{Epith}) varies across different OED grades and transformation outcomes. Boxplots illustrating these distributions are now included in the Results

section as a new figure, providing further insights into the variability of R_{Epith} and its potential prognostic relevance.

Furthermore, the Methods and Results sections have been updated to include this additional information and clearly describe the process used to determine and validate the threshold.

- 1.4 Pipeline Novelty: The novelty of the proposed pipeline is not well substantiated, given that previous state-of-the-art models already perform well. The paper's main strength is its evaluation on three external datasets. The discussion includes a statement that "These segmentations have been used in further DL models to predict grade or transformation or for pathologist-curated features based on digital images. However, there has been little focus on segmenting dysplastic regions solely for downstream prediction of malignant transformation." This is somewhat unclear, as the primary clinical outcome is to predict transformation. The statement does not fully justify the added segmentation steps.

We thank the reviewer for their invaluable feedback. In our work, we indeed apply Transformers to segment the dysplastic region in OED cases, which is novel in itself. However, we further use these segmentations in our models to provide a fully automated pipeline to both classify a case as OED or normal, and predict whether the OED case will transform to malignancy. This is the first study to fully integrate these steps into a complete pipeline, whilst evaluating it on three external datasets. We acknowledge the importance of clearly highlighting this contributions and have updated the Discussion section to address this point more explicitly. In particular, we emphasise that the segmentation of dysplastic regions is a critical component of the ODYN pipeline, enabling more accurate downstream classification and transformation prediction, which sets it apart from the existing state-of-the-art models.

- 1.5 Case Selection and Staining Clarification: The authors state, "After microscopic inspection of the tissue sections by a Consultant Pathologist (SAK), newly cut 4 μm sections" were used. Does this mean that diagnostic slides were not used for training, but instead fresh, newly stained sections were used? Could the authors clarify if these sections were stained simultaneously or at different times? If at different times, in batches of how many slides per batch?

We thank the reviewer for highlighting this important point and the opportunity to clarify. Newly cut 4 μm sections were obtained from the original formalin-fixed paraffin-embedded (FFPE) blocks from the Pathology archive. The original diagnostic slides for the Sheffield cohort were used exclusively to confirm the suitability of each case and its dysplasia diagnosis prior to inclusion in the study. Once confirmed, fresh sections were prepared and stained using standard H&E protocols. The sectioning

and staining processes were conducted in batches between 2020 and 2023, with an average of 100 slides per batch. These details have been added to the Methods section of the manuscript.

- 1.6 High Case Attrition Due to Artifacts: It is surprising that the initial cohort of 406 patients was reduced to 358 due to artifacts, resulting in a loss of over 11% of cases. For a translational study, this suggests a notable impact on clinical utility. The authors have done prior work on artifact detection; could such approaches be used here to reduce case attrition? Alternatively, could the authors report model accuracy in the presence of artifacts?

We thank the reviewer for their constructive feedback on case attrition. To clarify, the reduction from 406 patients to 358 WSIs was not solely due to artifacts but rather, a vast majority of the excluded cases (42 cases) were excluded during the initial case review because they did not meet the study's inclusion criteria. These exclusions were due to factors such as insufficient epithelial tissue, the presence of ulceration or overlying infection, or features indicative of conditions like clinical oral lichen planus (OLP) or HPV-related lesions, which are distinct entities with different behaviour. Only six cases (1.5% of the total cohort) were excluded after scanning due to technical issues such as poor staining quality, artefacts, or blurring that rendered them unsuitable for analysis. Exploring model performance on slides with artifacts is another promising direction to enhance real-world applicability.

We have updated the manuscript to clarify inclusion/exclusion criteria, discussed these points as limitations, and emphasized the need for future prospective validation in larger datasets.

- 1.7 Utilization of Vision Transformers and ROIs: Although the authors leverage vision transformers within their architecture, they mention feeding the models with regions of interest (ROIs) from a large cohort of 260 slides, with the epithelium delineated in 105 WSIs. This raises questions about the consistency of ROIs in the training data and its impact on model performance.

We thank the reviewer for their insightful feedback. To clarify, the ROIs used for training were derived from a subset of the total cohort, ensuring coverage of both normal and dysplastic regions:

- For normal cases, the entire epithelium was segmented in 105 WSIs.
- For OED cases, dysplastic regions were specifically annotated in 260 WSIs from Sheffield and 48 WSIs from the external cohorts (Birmingham, Belfast, Brazil).

These ROIs typically spanned the entire tissue section in a WSI, where possible. However, since dysplasia is often localised rather than evenly distributed along the entire epithelium, it was necessary to annotate specific dysplastic regions to train the segmentation model effectively. This ensures the model could reliably differentiate dysplasia from adjacent normal areas. While the entire epithelium was segmented in normal cases, dysplastic ROIs in OED cases were focused on regions with confirmed dysplasia to optimise the model's performance for dysplasia detection.

We hope this explanation addresses the reviewer's concern. To improve clarity, we have added further details to the Methods section regarding our annotation strategy.

- 1.8 Figure 1: In the Epithelium/Nuclear Masks image, no nuclear segmentation mask is visible. Is this correct?

We thank the reviewer for their feedback. This is indeed correct, we did not include it as it would have made the image too busy; however, we have now included a zoomed-in cut-out showing the nuclear segmentations in a part of the image.

- 1.9 Table S2 (ODYN-SA Model): The supplementary table lists a model, ODYN-SA, which is not referenced in the methods or main text. Could the authors clarify if this is accurate?

We thank the reviewer for spotting this inconsistency. On external validation, we further tested the effect of Stain Augmentation (SA) on ODYN segmentation performance. We have added some comments on this to the Results section: "Further, stain augmentation (ODYN-SA, in Supplementary Appendix, Table S2) did not improve model performance."

- 1.10 Inclusion of Control Cases: It would strengthen the study to include control cases or OED cases that did not transform, allowing evaluation of dysplasia detector performance in distinguishing non-transforming from transforming OED.

We thank the reviewer for their feedback. We would like to clarify that the dysplasia detector section of ODYN is evaluating on detecting dysplasia in all cases and controls. This includes normal controls (on internal testing alone) and both transforming and non-transforming dysplasia cases (on both internal and external testing).

We have adapted the manuscript to specify the controls are "non-dysplastic" controls in the Methods and Results section. Unfortunately, no non-dysplastic controls were available to us in the external validation cohorts. We have added this as a limitation of our study.

- 1.11 Heatmap Colormap Selection: Jet colormaps can introduce interpretive bias in heatmaps. I recommend the authors consider this and refer to guidelines here: <https://www.nature.com/articles/s41467-020-19160-7>

We thank the reviewer for pointing this out, we have changed our heatmap colour from jet to hot in Figure 1 and Figure 2.

Reviewer #2:

- 2.1 Shephard et al constructed an AI based model to discern histopathological dysplasia in oral epithelium and to link the findings to the chance of malignant transformation. This is an important piece of work, well executed. Since the authors provide the toll they used, independent parties can reproduce the results. The authors used different scanners to scan whole slide images and they used different cohorts from different pathology institutes, thereby correcting for possible institutional related HE features. They use the WHO criteria to grade dysplasia and to divide them into a three tier system of mild, moderate and severe dysplasia. Next to that, the WHO describes 28 features which are indicative for dysplasia. The weakness of the present WHO description of oral epithelial dysplasia lies in the fact that in the three tier system there is a strong emphasis on the cytonuclear atypia of the epithelium and the extent in which this is present in the level of the epithelium.

We thank the reviewer for their time and insightful feedback.

- 2.2 We now know that there are also cases with little cytonuclear changes but mostly architectural changes, that are as prone to develop malignancy as cases with more pronounced cytonuclear changes (Wils LJ, et al. The role of differentiated dysplasia in the prediction of malignant transformation of oral leukoplakia. *J Oral Pathol Med.* 2023 Nov;52(10):930-938. doi: 10.1111/jop.13483. Epub 2023 Sep 25. PMID: 37749621.; Brouns ER, et al. Oral leukoplakia classification and staging system with incorporation of differentiated dysplasia. *Oral Dis.* 2023 Oct;29(7):2667-2676. doi: 10.1111/odi.14295. Epub 2022 Jul 7. PMID: 35765231.). Could the authors further elude to this in the discussion? They find a malignant transformation rate of 40% in the low-risk cases using the ODYN score in the external cohort (fig 3). Could this be due to the fact that the model puts to much emphasis on the cytonuclear features of atypia?

We thank the reviewer for highlighting the importance of evaluating the ODYN-score's reliance on cytonuclear features. We would like to clarify that the malignant transformation rate for low-risk ODYN cases in the external cohort is 22%, not 40% as mentioned. However, the concern remains that architectural changes, which are not explicitly captured in the ODYN-score, may contribute significantly to malignant progression. We acknowledge this limitation and agree that incorporating architectural

dysplasia features into future iterations of ODYN could further enhance its prognostic accuracy. We have added these important points (with references) as future directions in the Discussion section.

Manuscript# COMMSMED-24-1120A

Development and Validation of an Artificial Intelligence-based Pipeline for Predicting Oral Epithelial Dysplasia Malignant Transformation

We express our gratitude to the reviewers and the editorial team for their time and thoughtful feedback on our manuscript. We have completed the required editorial checklist and included the requested Supplementary Data files, including source data.

While preparing the requested Supplementary Data files, we noticed that 80 out of ~350 cases had been inadvertently omitted from the survival analysis. We have now included these cases, leading to a slight increase in our model C-Indices and Hazard Ratios but no change in the overall conclusions or discussion. For transparency, we present both sets of results in the Table below, with changes highlighted in red.

Additionally, in incorporating the required “Statistics and Replication” section, we made minor adjustments to the ordering of the Methods section for clarity.

Table 1. Slide-level results for transformation prediction. Here, WHO grade G1 is mild vs moderate/severe cases, whilst WHO grade G2 is mild/moderate vs severe cases. For AUROC, AUPRC and C-Index, the mean value is given with the standard deviation in brackets. For the hazard ratio, HR, we additionally provide the 95% confidence interval in square brackets.

Model	Internal Validation					External Validation				
	AUROC	AUPRC	HR	p	C-Index	AUROC	AUPRC	HR	p	C-Index
WHO grade G1	0.67 (0.03)	0.63 (0.07)	9.16 [3.68 – 22.80] 10.93 [3.40 – 35.16]	< 0.001	0.66 0.67 (0.00)	0.65 (0.00)	0.72 (0.00)	2.43 [1.12 – 5.29]	0.025	0.61 (0.00)
WHO grade G2	0.61 (0.06)	0.47 (0.11)	2.71 [1.73 – 4.26] 2.10 [1.23 – 3.58]	< 0.001	0.62 0.60 (0.00)	0.57 (0.00)	0.59 (0.00)	1.62 [0.82 – 3.19]	0.164	0.56 (0.00)
Binary grade	0.73 (0.05)	0.62 (0.07)	6.03 [3.63 – 10.01] 4.95 [2.78 – 8.81]	< 0.001	0.71 0.69 (0.00)	0.68 (0.00)	0.72 (0.00)	2.84 [1.36 – 5.92]	0.005	0.62 (0.00)
ODYN-score	0.71 (0.07)	0.43 (0.12)	3.86 [2.04 – 7.69] 3.40 [1.72 – 7.67]	< 0.001	0.66 0.65 (0.02)	0.73 (0.05)	0.67 (0.05)	2.95 [1.44 – 6.02]	0.003	0.63 (0.04)

Best values in bold.